# Determination of Endangered Freshwater Fishes: Can Value Be Estimated?

**Jay Richard Stauffer, Jr.** [1,*] **and Raymond Paul Morgan II** [2]

1   Ecosystem Science and Management, Penn State University, University Park, PA 6139, USA;
    Honorary Research Associate, South African Institute of Aquatic Biodiversity, Makhanda 6139, South Africa
2   Appalachian Laboratory, University of Maryland, Center for Environmental Science, 301 Braddock Road,
    Frostburg, MD 21532, USA
*   Correspondence: vc5@psu.edu

**Abstract:** The determination of endangered species is problematic. If one considers a species to be ontological individuals, then if a species goes extinct, it is gone forever. The Brook Trout is used as an example of a "species" which may be comprised of several unique entities that warrant a specific status. In addition to determining the specific status, it is difficult to determine how to place a monetary value on endangered species that do not have a general appeal to the public (e.g., many bird species), a commercial value, no known medical properties (e.g., deep water sponges vs. cancer), or generate monies for recreation. Perhaps if we could identify the unique information carried by a particular species, we could place a value on that information and assess the monetary value of the information lost.

**Keywords:** endangered status; value of endangered species; subspecies; Brook Trout

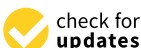



## 1. Introduction

Freshwater lakes and rivers comprise less than 0.0093% of the total water mass on Earth, yet over 41% of all species of fishes are found in freshwater systems [1]. Some species, e.g., the Longnose Sucker *Catostomus catostomus* (Forster), are widely distributed [2], while others are known from only a single locality (e.g., the Devils Hole Pupfish *Cyprinodon diabolis* Wales; [3]. Certainly, land is a more formidable barrier to freshwater fishes than water is for terrestrial vertebrates. Thus, many populations of fishes have been effectively isolated leading to allopatric speciation. As a result, intralacustrine allopatric speciation has been postulated to account for the rapid and extensive speciation by cichlid fishes in the Great Lakes of Africa [4–6]. Additionally, constraints within lotic systems (e.g., waterfalls, the limited stream habitat in small headwater streams, unsuitable water quality) have isolated populations of fishes, with Flebbe et al. [7] classifying these populations as veritable islands. Demographically, such populations have reduced effective population sizes and extinction may be accelerated [8]. Several authors have also suggested that sympatric speciation has occurred frequently in freshwater fishes [5,9].

The refutation of sympatric speciation revolves around the question of how reproductive isolation can arise prior to a physical barrier to gene flow [6]. Perhaps populations separate ecologically without geographical barriers by differences in habitat preference. For example, if a species is comprised of both gold morphs and blue-black morphs, perhaps the gold morphs are restricted to deeper water to avoid predation by surface predators. If such separation occurs, then assortative mating may result in a speciation event [10]. Such ecologically driven isolation may be caused by seasonal isolation [11], mate selection [12,13], and runaway sexual selection [14]. Alternative polymorphisms may also contribute to the extensive adaptive radiation and sympatric existence of closely related species [15].

Historically, research on biodiversity has centered on documenting the number of species through morphological and genetic data and identifying unique populations using

molecular data. This included direct observations and literature reviews of behavior, life history traits (e.g., reproductive modes, trophic level interactions, habitat utilization, etc.), and species interactions to provide essential data on the functional aspects of the taxa inhabiting selected systems. The assessment of ecosystems included species-area curves [16], diversity indices [17–20], autotrophic–heterotrophic ratios [21], saprobian designations [22,23], and biotic indices [21,24]. Ott [25] developed links between water quality indices and specific-use indices. Aquatic habitats have been classified based on calcium content [26], distribution of fauna [27,28], water zones [29], stream gradient [30], and stream order [31]. All these schemes for the assessment of ecosystems are anchored on the number of species in that system, the functional aspects of each species, and the genetic diversity and uniqueness of the system. In particular, the International Union for the Conservation of Nature (IUCN) emphasizes that a sound scientific basis is vital to conserving our biological resources. Their standards, which include IUCN Red List Categories and Criteria, Guidelines for Applying Protected Area Management Categories, and the Global Standard for Nature-Based Solutions, among others, are widely used to generate policies. Thus, decisions affecting endangered species must be based on current knowledge and evidence that is science-based.

## 2. Threats to Endangered Species

Barlow [32] discussed the threats of Earth's growing human population to the biodiversity of natural systems, and Diamond [33] discussed the "evil quartet" (i.e., 1—overharvesting, 2—habitat fragmentation and degradation, 3—exotic species, and 4—chains of extinction) that leads to a contemporary loss of biodiversity. The risks of burgeoning human populations and the associated pattern of diminishing resources on Earth's biota have all threatened biodiversity and endangered species. Certainly, one of the detrimental aspects of human populations is the decrease in genetic diversity connected to an increase in the rate of extinction of natural populations.

Probably the most drastic incident of mass extinction in our lifetimes occurred with the destruction of the haplochromine cichlid flock in Lake Victoria that was associated with the introduction of the Nile Perch *Lates niloticus* [34,35]. In addition to the loss of species due to introductions, the introduction of freshwater fishes has had major detrimental impacts on the native faunas of fishes throughout the world [36,37]. Besides exotic introductions, endangered species and overall biodiversity are threatened by the disruption of symbioses, species interactions, abiotic changes, community collapse, and responses to a myriad of anthropogenic stressors. Many of these stressors are easily recognized (e.g., industrial outfalls, removal of canopy, etc.), but others are more subtle, such as movement of natural populations across barriers and the introduction of the same "species".

*Brook Trout, Salvelinus Fontinalis, as a Case History*

The Brook Trout, *Salvelinus fontinalis* is native to headwater streams and cold-water lakes east of the Mississippi River from northeastern Canada, south through the Great Lakes, and into the southern Appalachian Mountains [38]. The geological history of this region reveals numerous events [39] that would serve to isolate fish populations, including extensive glacial impoundment and stream captures in the northern range, while not drastically affecting southern populations through glacier activity. Once highly abundant, Brook Trout are under increasing pressure from anthropomorphic changes in the landscape, pollution, competition from stocked non-native salmonids, and warming trends [40–43]. Understanding the interspecific and intraspecific population status of this species is critical to its continued viability and conservation efforts throughout its native range.

Behnke [44] posed the question of whether there are distinct northern and southern groups of Brook Trout, or a single homogenous stock established since the last glacial retreat during the Pleistocene, and furthermore stated that the Brook Trout is widely distributed and a more generalized species that is usually recognized as a single species. In contrast to Behnke's [44] statement, Burr and Mayden [45] countered by asserting that there are

many questions relative to the species-level taxonomy that are unresolved. Brook Trout truly represent a salmonid species where there are major questions as to genetic structure throughout its native range. Stauffer and King [46] distinguished populations that were sea run, from the North Atlantic Slope, from the St. Lawrence River and the Great Lakes drainages, from the upper interior basin, from the southern Atlantic Slope, and from the lower interior basin.

The distribution of Brook Trout in the eastern United States is now fragmented and compressed relative to the historical distribution for this species [43]. Three important processes, fragmentation, compression (defined as the forcing of Brook Trout into a limited and isolated headwater stream habitat), and extinction of populations are occurring as a result of increasing stressor effects. Fragmentation and compression reduce the effective population size of Brook Trout and accelerate extinction, or at the very least, lead to an increased extinction probability [8]. In addition, structural changes in stream hydrogeology and trophic dynamics, especially in rapidly urbanizing regions, may expedite these three processes [47]. Usually, compression of a population results from the presence of exotic species, in the lower section of a stream segment; isolation due to the presence of exotic species in nearby stream reaches; occurrence of a significant stressor (e.g., acid mine drainage) or multiple stressors; or the presence of physical barriers. Brook Trout in fragmented and compressed subpopulations are vulnerable to extinction in the near term (50–100 years).

Several authors have surveyed the genetic relationships of populations of Brook Trout throughout their native range as tests of competing biogeographic hypotheses. Stoneking et al. [48] surveyed populations from Tennessee and North Carolina and compared them with fish from New York and Pennsylvania using allozymes. Their results indicated the presence of separate northern and southern phylogenetic lineages to a level suggesting subspecific status. Quattro et al. [49] and Morgan and Baker [50] surveyed western Maryland populations of Brook Trout using mitochondrial DNA restriction fragment length polymorphisms (mtDNA RFLP) and allozymes, respectively, and found significant genetic divergence between populations from the Chesapeake Bay and Ohio River drainages. Additional allozyme evidence for the phylogeographic structure was found among populations from the Great Smoky Mountains [51,52], where significant genetic differentiation among native populations was detected. Hayes et al. [53] (1996) examined 11 native populations of Brook Trout from the southern Appalachian Mountains for mtDNA haplotypic diversity and found sequence divergence of up to 0.8% between populations.

A study from the southern edge of the glaciated region of eastern North America determined that there was a high degree of genetic variation in populations of Brook Trout from Pennsylvania and New York [54]. Most of this genetic diversity was partitioned among four major river basins (St. Lawrence, Delaware-Hudson, Susquehanna, and Allegheny). In recent studies, northern populations of Brook Trout, defined here as those from recently glaciated (i.e., Wisconsin glacial episode) regions in Canada and Great Lakes drainages in the United States, have been extensively characterized based on allozyme surveys and mtDNA RFLP [55–60]. A single mtDNA haplotype assemblage dominated most northern populations, indicating recolonization from a single Atlantic refugium [60]. Notable exceptions were several western Great Lakes populations that contained divergent mtDNA assemblages believed to have recolonized from a Mississippi River refugium, and divergent haplotypes in the Canadian Maritime Provinces thought to have reentered from an Acadian (northeastern coastal) refugium.

Besides demonstrating the evidence for the refugial origin of populations throughout the native range of Brook Trout, the study by Danzmann et al. [60] delineated six major Brook Trout clade assemblages. Morgan and Danzmann [61,62] and Hall et al. [63] suggest high mtDNA RFLP diversity among populations of Brook Trout in the mid-Atlantic when compared to northern populations analyzed previously [58,60]. These studies revealed the low mtDNA assemblage diversity in recently glaciated regions in Canada and northeastern USA.

Using microsatellite analyses, Morgan et al. [64] segregated Brook Trout from Maryland into five discrete geographical units—Upper Potomac River drainage, Catoctin Mountains, Patapsco/Gunpowder River drainages, Susquehanna River drainage, and Ohio River drainage (Youghiogheny River)—through both POPTREEW clustering [65] and STRUCTURE analyses [66]. Initially, it was determined that bootstrapping efficiencies were low in the phenogram but improved significantly once ten Brook Trout collections were removed from the clustering analyses (see Table 3 and Figure 6 in Morgan et al. [64]). Basically, these ten populations were either highly fragmented or disconnected (dam effects, acid mine drainage, etc.) from other populations and displayed rapid genetic drift in a relatively short time span (less than 100 years). In some cases, the effective population size was small and would greatly drive genetic drift. Thus, highly isolated populations of Brook Trout may need special management. Similarly, dam effects and acid mine drainage appear to have driven genetic diversity in Pennsylvanian Brook Trout from the Allegheny and Susquehanna basins through habitat fragmentation [67].

Kazyak et al. [68] completed an extensive survey of 818 populations of wild Appalachian Brook Trout. This study primarily focused on the southern Appalachians, defined as Maryland to Georgia. It was determined that fine scale population structuring was present, along with the deep genetic structure present over broad spatial extents, consistent with the Mississippian, mid-Atlantic, and Acadian glacial refugia for Brook Trout. Southern populations of Brook Trout tended to have small effective population sizes and that genetic drift was postulated to be a strong driver of current population structure. Genetic relationships of Brook Trout across the landscape appeared to be far more complex that what was earlier suggested [48], although the major clusters observed reflected dispersal events from three refugia [68]. In addition, it was suggested that assessing the adaptive potential in Brook Trout may increase the ability to optimize genetic management.

Most population studies from the southern Appalachians have produced evidence that genetic diversity is relatively high in the southern portion of the native range of Brook Trout [51–53]. The only population included from the southern range in Danzmann et al. [60] was Indian Camp Creek in the Tennessee River drainage. The single haplotype found in this population was the only haplotype comprising the most divergent assemblage from that study. This, coupled with the high mtDNA sequence divergences (as high as 0.8%) found between southern populations by Hayes et al. [53], suggest that regions south of the putative mid-Atlantic possess higher mtDNA haplotypic diversity. Based on these comparisons, however, mid-Atlantic populations emerged as transitional in an ascending continuum of haplotypic diversity from north to south. Future research should focus on this region.

The question arises in this discussion as to whether past stocking events followed by the naturalization of stocked fish or their hybridization with wild fish could account for these assemblage transfers between major drainages. One problem, in examining the genetic structure of populations from Maryland, is the rather large-scale introductions of eggs, fingerlings, and adults throughout the state in the early 1900s in response to anglers' concerns about the lack of trout, and other game species [69,70]. At that time, agriculture, timber harvesting, and mining had severely affected Maryland's trout streams. Only headwater streams (especially the upper reaches) were sampled in the Maryland work [61–64]. These headwater streams were generally not stocked due to early, primitive road systems that prevented stocking truck access. Initially, the stocking emphasis was on fingerlings but later was switched to stocking larger fish for put-and-take fisheries [70]. Fingerling survival appeared to be poor and angling pressure in the put-and-take areas quickly removed the larger Brook Trout [70]. Some Brook Trout from federal hatcheries were also stocked, but these were also larger fish placed into put-and-take zones. It should be noted that there was no strong evidence of any past stocking effects on Maryland Brook Trout [64]. Kazyak et al. [68] observed limited effects of hatchery stocking, but introgression did not affect the overall broad-scale signal pattern of genetic structure. In a study of rear edge populations of Brook Trout in South Carolina, Pregler et al. [71] found effects of

hatchery introductions in 6 of 18 streams sampled, and postulated management options for enhancing or restoring these rear edge populations.

The current work focuses on analyses of microsatellite loci in Brook Trout, demonstrating the power of this bioanalytical technique in understanding the population structure of native Brook Trout throughout its range. Brook Trout is the only salmonid native to the Southern Appalachians and functions as a keystone species in some headwater streams. Since the turn of the century, this native fish has lost approximately 75 percent of its range in Great Smoky Mountain National Park (GRSM). The historic use of hatchery-reared Brook Trout for supplemental and restorative stocking in GRSM, and the potential negative impacts to the overall health and long-term maintenance of populations resulting from this management approach, underscore the need to recognize the lineage (evolutionary trajectory) of each stream population.

Genetic variation detected at 13 microsatellite DNA loci in over 700 GRSM Brook Trout has shown a range of allelic diversity (including some of the lowest levels reported for vertebrate populations). Statistical analyses (AMOVA, using pairwise genetic distances—[72] have shown a high degree of genetic differentiation among the three drainages and among tributaries within the drainages. Genetic distances ($F'_{ST}$) among all the collections are large, suggesting a high degree of reproductive isolation exists among and within streams in GRSM streams. Genetic distances observed among (0.65–0.77) and within drainages (0.45–0.57) are like those observed among some accepted subspecies of fishes (e.g., North American and European Atlantic salmon, *Salmo salar,* and Atlantic and Gulf sturgeon, *Acipenser* spp. respectively). Similarly, an analysis of molecular variance indicates that 18.0% of the total genetic variance was observed among Brook Trout inhabiting the three river drainages, 19.2% was observed within the drainages, and the remaining 62.8% can be attributed to within collection variation. This indicates the presence of highly significant differentiation at all hieratical levels, which suggests that the individual stream should be considered the unit of management. Maximum likelihood assignment tests confirm this assertion as the correct assignment of fish to each collection averaged 98.5%, and assignment to drainage was 100%.

These findings create somewhat of a management conundrum for GRSM. Does the genetic divergence observed among GRSM Brook Trout show adaptive significance or is the variation due to neutral drift? Stauffer [73] described three species of *Salvelinus* from the Pigeon/French Broad drainage. When stocked together into LeConte Creek near Gatlinburg, TN in 1999, they each successfully established a reproducing population. Fin clips sampled from Brook Trout from LeConte Creek in 2006, and genotyped at 13 microsatellite loci, suggested that fish from the three distinct streams did not mate randomly but exhibited positive (selective) assortative mating. This behavior, when sexually reproducing organisms tend to mate with individuals that are like themselves, can have the effect of reducing the range of variation, when the assorting is cued on heritable traits. If the assortment is based on genotype, the dynamics at a single locus are similar to those under inbreeding conditions. The only difference would be that inbreeding affects all loci equally, whereas assortative mating acts primarily on the few loci governing a phenotypic trait(s). This behavior, if confirmed, could have a significant impact on future supplementation and restoration plans for Brook Trout throughout the species range.

The independent evolution of adaptively divergent phenotypes among closely related lineages is most likely the consequence of natural selection. Because natural selection ultimately acts on the genetic variation underlying character variation, identifying the genes associated with parallel evolutionary changes among recently diverged lineages is essential to uncovering candidate genes implicated in adaptive phenotypic variance. Contemporary genomic approaches can be particularly useful for identifying genetic targets of selection and genetic mechanisms of adaptation even among recently diverged lineages. For example, comparative genomic expression profiles generated by cDNA microarrays (or gene chips) were a major first step in identifying an initial pool of candidate genes that might be implicated in adaptation, particularly in non-model systems for which whole

genomes sequences are unavailable. Gene control elements (e.g., regulatory genes) are likely an important source of adaptive variation. DNA microarrays are a powerful method for the global analysis of steady-state intracellular mRNA levels, and thus identifying genes that are transcriptionally modulated because of metabolic or bioenergetic demands. The information gathered from these arrays of gene sequences can be used to characterize complex biological processes and interactions. Specialized expression arrays containing genes related by function, tissue, or pathway are becoming widely adopted by researchers studying the adaptive significance of genetic differentiation. Given that few gene sequences are available for Brook Trout, an analysis of global gene regulation may be the most direct avenue to identifying diagnosable genetic polymorphisms between genetically divergent populations occurring within watersheds and to understanding the evolutionary differences that exist between recently diverged lineages.

Mid-Atlantic Brook Trout are under increasing pressure from urbanization and pollution such as acid mine drainage [74,75]. In addition, Brook Trout may disappear from low altitude regions, such as those on the Piedmont of the mid-Atlantic, due to climatic warming [76]. Population models indicate that multiple anthropogenic stresses have the potential to cause serious population declines [77]. Though the Brook Trout is currently considered stable throughout most of its native range [75], certain populations may become increasingly rare in the future, especially those near growing urban areas.

The transitional status of mid-Atlantic populations of Brook Trout provides an opportunity to conserve a significant amount of genetic diversity within a relatively small area. The natural history (i.e., stream capture, historical isolation, etc.) of the streams sampled from this region is likely to have made a greater contribution to extant assemblage structure than anthropomorphic actions (i.e., stocking). Therefore, management of Brook Trout in the mid-Atlantic should be based on major hydrological divisions that separate major lineages except where evidence exists for assemblage mixing by natural means. In these drainages, more subtle management divisions are warranted.

There is increasing interest in protecting and restoring populations of Brook Trout within their native range. Prior to any recovery, restoration, or protection activity or effort, it is critical, from a conservation genetics perspective, to understand the genetic structure, or variation, of the species of concern. Understanding genetic variation is simply a tool that may drive the identification of the appropriate management or conservation unit, acting in both evolutionary and ecological processes. For example, the National Park Service Inventory Program System, in order to meet certain legal and congressional mandates, must assess biotic components within its parks. Many eastern NPS parks have significant populations of Brook Trout that are currently being assessed for both genetic and population structure. In addition to the NPS, the International Association of Fish and Wildlife Agencies, with major direction and support from the USFWS, are actively developing a collaborative effort, among the eastern states with Brook Trout, to restore, enhance, and protect these populations. The Eastern Brook Trout Joint Venture is the nation's first pilot project under the National Fish Habitat Initiative, directing locally driven efforts that build private and public partnerships to improve fish habitat. Long-term goals are to develop a comprehensive restoration and education strategy to improve the aquatic habitat; to raise education awareness; and to raise federal, state, and local funds for the conservation of Brook Trout.

States are also active in the conservation of Brook Trout. Currently, Maryland now has a statewide management plan for Brook Trout in 2006: the first plan for a freshwater species [78]. New York identified, through genetic work, its heritage populations of Brook Trout, and instituted policies to protect these fishes. New Jersey and North Carolina are completing inventories of their Brook Trout, with the intent to develop management plans. Trout Unlimited is addressing Brook Trout protection and restoration through its New England Brook Trout and "Bring Back the Natives" initiatives.

Hocutt et al. [39] hypothesized that stream captures account for the low fish community diversity of tributaries of the Chesapeake Bay between the Susquehanna River and the

Potomac River. If true, stream captures could also account for the admixture of mtDNA assemblages in this area. During the Pleistocene, stream captures by one or more tributaries to the Susquehanna just north of the present-day Maryland border could have "fixed" this haplotype in certain areas. An alternative hypothesis is that the Susquehanna River served as a dispersal route connecting all presently separated Chesapeake Bay drainages [79].

Hocutt et al. [39] recognized that the Savage River (Chesapeake Bay Drainage) in western Maryland captured roughly 207 km$^2$ of the upper Casselman River (Ohio-Mississippi drainage). Howard and Morgan [80] tested the hypothesis of a Susquehanna dispersal route for mottled sculpin *Cottus bairdi* and found evidence for faunal transfer between these drainages. Big Piney Branch, a tributary of the Casselman River in the Youghiogheny drainage, contains fauna not common to the rest of the western Maryland Ohio-Mississippi drainage. This may indicate a reverse transfer of fish to the Youghiogheny drainage. Hubbs and Lagler [81] postulated the transfer of blacknose dace, *Rhinichthys atratulus* via this route.

## 3. Implications for Management and Conservation

Certainly, local populations are sometimes unique and should be protected. For example, there are many populations of Burbot (Lota lota) that are separated from the main populations that may be undescribed species. The Allegheny Burbot (*Lota* spp.) occurs in the smaller streams of the upper Allegheny River in Pennsylvania [82]. It differs from the Burbot found in Lake Erie by being slenderer, having a larger eye, and fewer dorsal-fin and anal-fin rays [82]. Additionally, there is a population of smaller Burbot in the Susquehanna River. Do these populations deserve special protection? Similarly, the Blackchin Shiner (*Notropis heterodon*) is widely distributed in the Great Lakes and upper Mississippi River, but in Pennsylvania it is confined to small lakes in Erie County [82]. Again, since these populations are isolated, should they be protected?

As stated above, the populations of mid-Atlantic Brook Trout are under increasing pressure from urbanization and pollution such as acid mine drainage [74,75]. As indicated above, there may be several species of "Brook Trout" within the overall range, many of which may be endangered. Certainly, there are unique populations of many fishes (e.g., Longnose Sucker), which are not designated as species, but are unique in their genetic structure and distribution. The question then becomes: what is the value of an endangered species/population that is extremely isolated, has no real commercial value, and is pretty much unknown? When Etnier [83] discovered the snail darter, *Etheostoma tanasi*, in the Little Tennessee River, it halted the construction of the Tellico Dam. As indicated earlier, if one views species as ontological individuals, then when they are lost, they are gone forever. Perhaps if we could identify the unique information carried by a particular species, we could place a value on that information and assess the monetary value of the information lost. As a society, we need to decide as to whether this information is worth saving.

Canessa et al. [84] provided a primer on calculating the value of information for applied ecologists and concluded that the value of information is dependent on current knowledge, quality of information, and expected outcomes of available management actions. Spacy [85] defined information as data that interpreted by humans become knowledge and listed some 20 types of information. In business, the expected value of information is certainly dependent upon the probability of the event (Expected value of perfect information—Wikipedia). In his excellent book entitled Wonderful Life, Gould [86], when discussing replaying the tape of life, gives the example that the combinations of 10 items from a pool of 100 yields more than 17 trillion potential outcomes. Therefore, the evolution of the same species following extinction approaches zero, or as Captain Corcoran stated, "hardly ever" [87]. Thus, if we assume that the information lost if a species goes extinct is not retrievable, then the value of that information is off the charts. In effect, it is comparable to uncertainty in business, where uncertainty is a certainty, and therefore information is cost intensive [88] (Thakur Why Management Information Systems (MIS) Are Required?—Computer Notes (ecomputernotes.com, accessed on 6 July 2022).

Before the decision of worth saving versus not worth saving is reached, we have to put a value on the information (VoI). Wilson [89] described VoI as a way to estimate the value of an expected gain by reducing uncertainty though some form of the collection of data. Obviously, this is difficult since the value of information per se does not have any universal value [89]. If species lack the general appeal to the public (e.g., many bird species), a commercial value, does not have any known medical properties (e.g., deep water sponges vs. cancer), or generate monies for recreation it becomes more difficult to give them a value and enter them into a benefit/cost analysis. Bartkowski et al. [90] stated that when valuating biodiversity, one should use multi-attribute approaches that emphasize the roles biodiversity play for human well-being. When placing a cost on the extinction of a species, it must be realized that the outcome of extinction is irreversible [91]. This irreversibility places a premium on decisions that maintain flexibility, and if the potential value of future information is ignored, then the worth of a species will be undervalued [91].

Norton [92] recognized that some species have a commodity value (it can be bought or sold), an amenity value if its existence improves our lives in a nonmaterial way (e.g., joy at seeing a hummingbird), and a moral value (i.e., valuable in themselves and not dependent on value to humans). With respect to endangered species, perhaps we should measure the amount of unique information that would be lost with extinction. We could measure the amount of information, but it would be extremely difficult to estimate the payoff that would be lost for species that do not have a commercial value. In business, the payoff could include profit, output, or revenue, but there are also less tangible ones such as happiness, welfare, or utility [88]. Certainly, the protection and preservation of a unique population/species of Brook Trout would have some unknown intrinsic value that some would consider worth saving while others would not. The payoff could be estimated by revenue generated by those that utilize the population in a recreational context, but the payoff may be much greater than that if some aspect of the genome has some, as of now unknown, important information relative to the preservation of biodiversity or human health. In conclusion, perhaps it is impossible to assign a value to endangered fishes or any species. Ehrenfeld [93] stated: "Assigning value to that which we do not own and whose purpose we cannot understand except in the most superficial ways is the ultimate in presumptuous folly".

**Author Contributions:** Conceptualization, J.R.S.J.; writing—original draft preparation J.R.S.J.; writing—review and editing J.R.S.J. and R.P.M.II. All authors have read and agreed to the published version of the manuscript.

**Funding:** We appreciated partial support for this project from the USDA National Institute of Food and Agriculture, under Hatch project #PEN04584 (J.R.S.J.). Any opinions, findings, and conclusions or recommendations expressed in this material are those of the authors and do not necessarily reflect the views of the funding agency.

**Data Availability Statement:** Not applicable.

**Acknowledgments:** We benefitted from discussions with the late Timothy L. King.

**Conflicts of Interest:** The authors declare no conflict of interest.

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
