# Peer review of "Determination of Endangered Freshwater Fishes: Can Value Be Estimated?"

_water, doi:10.3390/w14162524_

Round 1
Reviewer 1 Report
The authors provide a fresh and very timely perspective on the potential development of practical conservation strategies. Determining a set of unique information characteristics on a species group such as the highly visible trout common in many streams could provide increased public support for conservation efforts. Characteristics that showcase the economic and aesthetic value of a 'representative fish species' as part of the overall health of a stream or lake can bolster support for conservation efforts by refining and clarifying the general concept of ecosystem services. The concepts offered in the manuscript will provide a very significant contribution to the current literature on the conservation of natural capital.
Author Response
I completed a spell check.
Reviewer 2 Report
This is an interesting commentary that is of wide relevance to the conservation of freshwater fishes generally. The script is well prepared but still requires a careful edit in order to eliminate minor typos, the odd spelling (line 62 Endangered, 239 hierarchical), and omissions (line 336 ?). There are inconsistencies in the citation of certain references (italics vs no italics in cited journals, page numbers for some books and not others etc).
My major concern is that no reference is made to the IUCN evaluation of threatened species that is the general system followed for evaluating species. For this commentary to be generally relevant it needs to consider the system used by the world at large and place that in the context of the given examples. Management considerations generally follow, or are facilitated by, such a system outcomes.
Author Response
I added a paragraph at the end of the introduction that discusses the role of IUCN.
Round 2
Reviewer 2 Report
Inclusion of reference to the IUCN system as suggested, has been done. Other minor corrections are made. The commentary is thorough and provides a useful addition to the literature when consideration of threatened fishes (and other organisms) is under scrutiny.
This manuscript is a resubmission of an earlier submission. The following is a list of the peer review reports and author responses from that submission.
Round 1
Reviewer 1 Report
I started to read this piece of work with great interest, since it promised to provide insights into the determination and cost of endangered freshwater fishes. However, after going through the first more general introductory sections, it become clear that the main focus of this manuscript is revising already published population genetic studies on Brook Trout without returning to the main issue and offering potential solutions. As such, it become rather challenging to link the information provided on Brook Trout with determination and cost of endangered species. Therefore, I think the authors have missed the opportunity to more clearly define or summarize critical issues associated with determination of endangered species and how to put a price tag on it. Therefore, I suggest that the authors modify both the title, abstract and main text to be focus more specifically on Brook trout population genetics (and also taking more forward looking view on new technologies) or alternatively, rewrite the manuscript placing much more emphasis on important issues raised in the title and abstract.
Other comments and suggestions
1) Is the longnose sucker truly panmictic species? In a panmictic species, all of the individuals of a single species are potential partners, and the species gives no mating restrictions throughout the population. Although I am not the expert on this species ecology, it is hard to imagine that the longnose sucker is truly panmictic species given that it is distributed in north from the Columbia, Delaware, Missouri and Monongahela river basins, as well as the Great Lakes basin. Based on work by Langille et al. (2016) for example, longnose sucker hardly can be considered as panmictic.
2) The majority of citations are notably rather old. So this arises question if more recent studies do not add anything substantial for what we know already or if the reference list is incomplete. For example, the section (lines 263-278) describes cDNA microarrays as one of the cutting edge genomic methodologies. I think this is a bit outdated since in addition to gene expression arrays, an increasing number of researchers are using RNAseq, long noncoding RNAs (lncRNAs), epigenetic assays (methylation, ATACseq etc) and other omic approaches to understand how gene expression is regulated at transcriptional, RNA processing, translational, and post‐translational levels. Furthermore, other technological developments, such as long read sequencing, whole genome assembly approaches, SNP genotyping arrays, whole genome association approached could be mentioned if one wants to provide more comprehensive and updated overview about recent technological developments relevant for conservation and restoration.
3) Line 230: Please specify which genetic distance you refer to.
4) Lines 339-355: I am missing here more general and comprehensive discussion about the implications of population structuring for management and conservation as currently, this section is very brief.
5) Lines 352-355: The authors state in the abstract and at the end of the manuscript that "Perhaps if we could identify the unique information carried by a particular species, we could place a value on that information and assess the monetary value of information lost" but provide no insights on how to place the monetary value of potentially lost information or ecological function. Therefore, if this point is to be raised, the authors need to put much more emphasis on describing this approach.
Langille, Perry, Keefe, Barker, Marshall (2016) Mitochondrial population structure and post-glacial dispersal of longnose sucker Catostomus catostomus in Labrador, Canada: evidence for multiple refugial origins and limited ongoing gene flow. Comparative Study J Fish Biol., 89(2):1378-1392. doi: 10.1111/jfb.13042.
Reviewer 2 Report
The article deals with a systematic review on the subject, presenting the proposed points in a very clear and satisfactory way. I believe that the topic is relevant for the conservation of aquatic biodiversity, in addition to integrating the use of genetic diversity and conservation. Complicated but converging topics.
Reviewer 3 Report
The content of the title and abstract are not related and discussed in the central text of the bibliographic review